# Cellular resolution models for *even skipped* regulation in the entire *Drosophila* embryo

**Garth R Ilsley[1,2]\*, Jasmin Fisher[3,4], Rolf Apweiler[1], Angela H DePace[5†], Nicholas M Luscombe[1,2,6,7†]**

[1]European Molecular Biology Laboratory, European Bioinformatics Institute, Wellcome Trust Genome Campus, Cambridge, United Kingdom; [2]Okinawa Institute of Science and Technology Graduate University, Okinawa, Japan; [3]Microsoft Research Cambridge, Cambridge, United Kingdom; [4]Department of Biochemistry, University of Cambridge, Cambridge, United Kingdom; [5]Department of Systems Biology, Harvard Medical School, Boston, United States; [6]UCL Genetics Institute, Department of Genetics, Evolution, and Environment, University College London, London, United Kingdom; [7]London Research Institute, Cancer Research UK, London, United Kingdom

**Abstract** Transcriptional control ensures genes are expressed in the right amounts at the correct times and locations. Understanding quantitatively how regulatory systems convert input signals to appropriate outputs remains a challenge. For the first time, we successfully model *even skipped* (*eve*) stripes 2 and 3+7 across the entire fly embryo at cellular resolution. A straightforward statistical relationship explains how transcription factor (TF) concentrations define *eve's* complex spatial expression, without the need for pairwise interactions or cross-regulatory dynamics. Simulating thousands of TF combinations, we recover known regulators and suggest new candidates. Finally, we accurately predict the intricate effects of perturbations including TF mutations and misexpression. Our approach imposes minimal assumptions about regulatory function; instead we infer underlying mechanisms from models that best fit the data, like the lack of TF-specific thresholds and the positional value of homotypic interactions. Our study provides a general and quantitative method for elucidating the regulation of diverse biological systems.

**\*For correspondence:** garth.ilsley@oist.jp

†These authors contributed equally to this work

**Competing interests:** The authors declare that no competing interests exist.

**Reviewing editor**: Roderic Guigo, Center for Genomic Regulation, Spain

## Introduction

A detailed knowledge of transcriptional control will have profound consequences for our understanding of myriad biological processes, including development, homeostasis, and evolution of new phenotypes. To this end, through a combination of genomic, genetic, and molecular experiments, the field continues to accumulate considerable information documenting components of regulatory systems and regulator-target interactions (*Gerstein et al., 2010*; *The modENCODE Consortium, 2010*; *The ENCODE Project Consortium, 2012*). At present however, many of these descriptions are qualitative. A major goal going forward is to interpret these data in a quantitative manner (*Wilczynski and Furlong, 2010*): how do regulators and regulatory interactions convert input signals to the appropriate output expression pattern? In general, answering these questions remains a significant challenge. The experiments needed to probe regulatory functions in detail are technically demanding; moreover, many systems involve multiple layers of control that cannot be investigated within a single experimental set-up. Theoretical models can help advance experimental investigations by providing a framework for deriving general principles and developing testable hypotheses (*Reeves et al., 2006*; *Tomlin and Axelrod, 2007*; *Lewis, 2008*; *Oates et al., 2009*; *Davidson, 2010*). An effective model should be able to define and predict expression accurately by describing how and by how much regulators influence target gene expression (*Hasty et al., 2001*; *Segal and Widom, 2009*).

**eLife digest** The transcription of genes into messenger RNA (mRNA) molecules is one of the most important processes in biology, but our present understanding of this process is largely qualitative. Molecules such as transcription factors and regions of DNA other than the region that codes for the mRNA are known to interact with each other to influence the onset of transcription, and also the rate at which it occurs. However, given the cellular concentrations of transcription factors in a developing organism, it is not known if it is possible to accurately predict their effects on transcription. Being able to make such predictions would greatly improve our understanding of how transcription and the development of an organism are controlled.

Ilsley et al. have tackled this problem by analysing a large volume of data called the Virtual Embryo dataset: produced by the Berkeley *Drosophila* Transcription Network Project, this dataset includes the results of mRNA expression measurements on 95 different genes at six different times during the early development of *Drosophila melanogaster*, a species of fruit fly. In particular, Ilsley et al. focussed on the expression at one point in time of the *even skipped (eve)* gene, a widely studied gene that is important for embryo development in these fruit flies. The *eve* gene is one of the genes responsible for dividing the fly into segments which form part of its body plan.

Without making any assumptions about the biological mechanisms that might be involved, Ilsley et al. built a statistical model that was able to predict the pattern of gene expression for a fruit fly, given the concentrations of the relevant transcription factors in the various cells within the embryo as input. The model was also able to predict the patterns of gene expression observed in other experiments involving mutations and the misexpression of fruit fly genes. Moreover, Ilsley et al. have made various predictions involving the genes Bicoid and Hunchback that can be tested experimentally in future studies.

Transcription in animals is controlled by interaction among transcription factors (TFs), enhancers, core promoters, silencers, insulators, and chromatin structure (*Lemon and Tjian, 2000*; *Arnosti, 2003*; *Levine, 2010*; *Ohler and Wassarman, 2010*; *Dean, 2011*). It is thought that core promoter elements and chromatin structure provide general competence for transcription at transcription start sites (*Lenhard et al., 2012*), whereas more distant enhancers up-regulate expression of genes under specific conditions (*Bulger and Groudine, 2011*; *Ong and Corces, 2011*). A single gene can be regulated by multiple enhancers, each directing a portion of the overall gene expression pattern in space and time. Enhancers operate by binding TFs, which in turn recruit regulatory co-factors and/or interact directly with the core promoter where RNA polymerase acts (*Spitz and Furlong, 2012*). A comprehensive model of transcriptional regulation would therefore include many factors, such as regulatory DNA sequence, DNA conformation, TF concentrations and nucleosome position among others (*Segal and Widom, 2009*). However, many of the parameters in such a model are currently impossible to measure. In the absence of such measurements, a partial yet predictive model based on available data is still valuable.

Here, we propose models of transcriptional control that are highly predictive of target gene expression given only TF concentrations at cellular resolution. Our goal is to make few assumptions about the underlying molecular mechanism. Instead, by generating models that predict experimental measurements as accurately as possible, we infer probable biological mechanisms and insights suggested by the parameters of the models. To achieve this, we focus on modeling the functional link between TF inputs and the resulting output (i.e., the 'regulatory input function'). These models are specific to individual enhancers: they capture how genomic loci interpret TF concentrations to control the output expression level of their target genes. Though multiple previous modeling studies have explicitly included protein–DNA interactions (e.g., in *Drosophila*, see *He et al., 2010*; *Janssens et al., 2006*; *Junion et al., 2012*; *Kazemian et al., 2010*; *Segal et al., 2008*; *Zinzen et al., 2009*), here, we choose to model the relationship between inputs and outputs directly as this offers several advantages. First and most importantly, this type of model encapsulates numerous relevant levels of biophysical interactions (i.e., TF-DNA, TF-TF, enhancer-promoter etc). Second, it enables us evaluate the utility of higher-order interactions between TFs, propose potential regulators and consider alternative hypotheses of experimental results. Third, in the context of developmental

biology, it allows us to explore the minimal information required to define positional information in the early embryo. Finally, focusing on input and output measurements means that the approach is applicable to relatively uncharacterized systems, for instance where enhancer regions have not yet been identified, or in assessing the conservation of regulatory input functions between species (*Wunderlich et al., 2012*).

We develop and test our models in the context of the well-studied *even skipped* (*eve*) enhancers in order to demonstrate their accuracy and utility. *eve* is expressed in a symmetrical pattern of seven stripes that subdivide the embryo along the anteroposterior axis (*Nüsslein-Volhard and Wieschaus, 1980*). Each stripe is only a few nuclei wide and any regulatory input function of an enhancer must define at least two borders at a high level of precision. A number of well-characterized enhancers direct expression of the seven *eve* stripes individually or in pairs (*Goto et al., 1989*; *Harding et al., 1989*; *Fujioka et al., 1999*). Here, we focus on the enhancers *eve 2* and *eve 3+7*, which have been shown to control stripe 2 and stripes 3 and 7 respectively (*Goto et al., 1989*; *Harding et al., 1989*; *Stanojevic et al., 1991*; *Small et al., 1992*, *1996*). Many of the input TFs and their roles in regulating *eve* expression have been defined; however, there remain unexplained properties underlying their regulation. An advantage of modeling *eve* is that we can use the available information as independent validations of our ability to recover known regulators and predict the outcome of regulatory perturbations, while also producing new insights. It is notable that though there has been some success in simulating the simpler gap gene expression pattern and in predicting *eve* expression on a portion of the anteroposterior axis, modeling pair-rule expression accurately across the whole embryo has remained a significant challenge (*Jaeger et al., 2004*; *Janssens et al., 2006*; *Papatsenko and Levine, 2008*; *Segal et al., 2008*; *Kazemian et al., 2010*; *Kim et al., 2013*).

To fit regulatory input functions, we require accurate measurements of expression levels for both the regulating TFs and *eve*. The Virtual Embryo from the Berkeley *Drosophila* Transcription Network Project provides the best available data for this purpose (*Fowlkes et al., 2008*). It is a cellular resolution, spatiotemporal atlas of gene expression and morphology for a whole *Drosophila melanogaster* blastoderm embryo. The dataset contains the three-dimensional coordinates for 6078 nuclei, along with mRNA expression measurements of 95 different genes at six time points during the 50 min leading to gastrulation: these genes include critical TFs that direct patterning in the early *Drosophila* embryo.

Using our modeling framework, we (i) predict expression patterns with accuracy and explanatory power at cellular resolution across the whole embryo; (ii) recover previously described regulatory relationships and test whether they provide sufficient positional information to define the resulting expression pattern; (iii) propose potential new regulatory relationships by comparing alternative models; and (iv) predict expression patterns under perturbation of input TFs, capturing the outcome of knockdown and misexpression experiments. Given the high level of accuracy of our models, we conclude with observations regarding mechanism and principles of enhancer function.

## Results

### Approach of this study

Our strategy is to find the logistic regression coefficients that most accurately describe the relationship between measured regulator concentrations and specific stripes of *eve* expression (*Figure 1*). We first train our models using the known regulators described in the literature to evaluate if they are sufficient for determining *eve* expression. At this stage, we also test the model for consistency across different subsets of the data. Next, we ask generally which regulators are able to specify *eve* expression (regulator discovery) and consider the plausibility of concentration-dependent dual regulation. Finally, we assess whether our models are able to predict beyond the conditions of the training data: specifically, we test whether our models can predict expression under perturbation, such as mutation of TFs and their cognate enhancer binding sites, by comparing our predictions with independently published experimental results.

### A classification approach for modeling *eve* expression

#### Preparing the data

We trained our models on a single time point: this has the advantage of eliminating uncertainties regarding nuclei assignments across time points in the Virtual Embryo dataset, yet still provides sufficient data for fitting. All models were trained using the third time point corresponding to ~30 min before

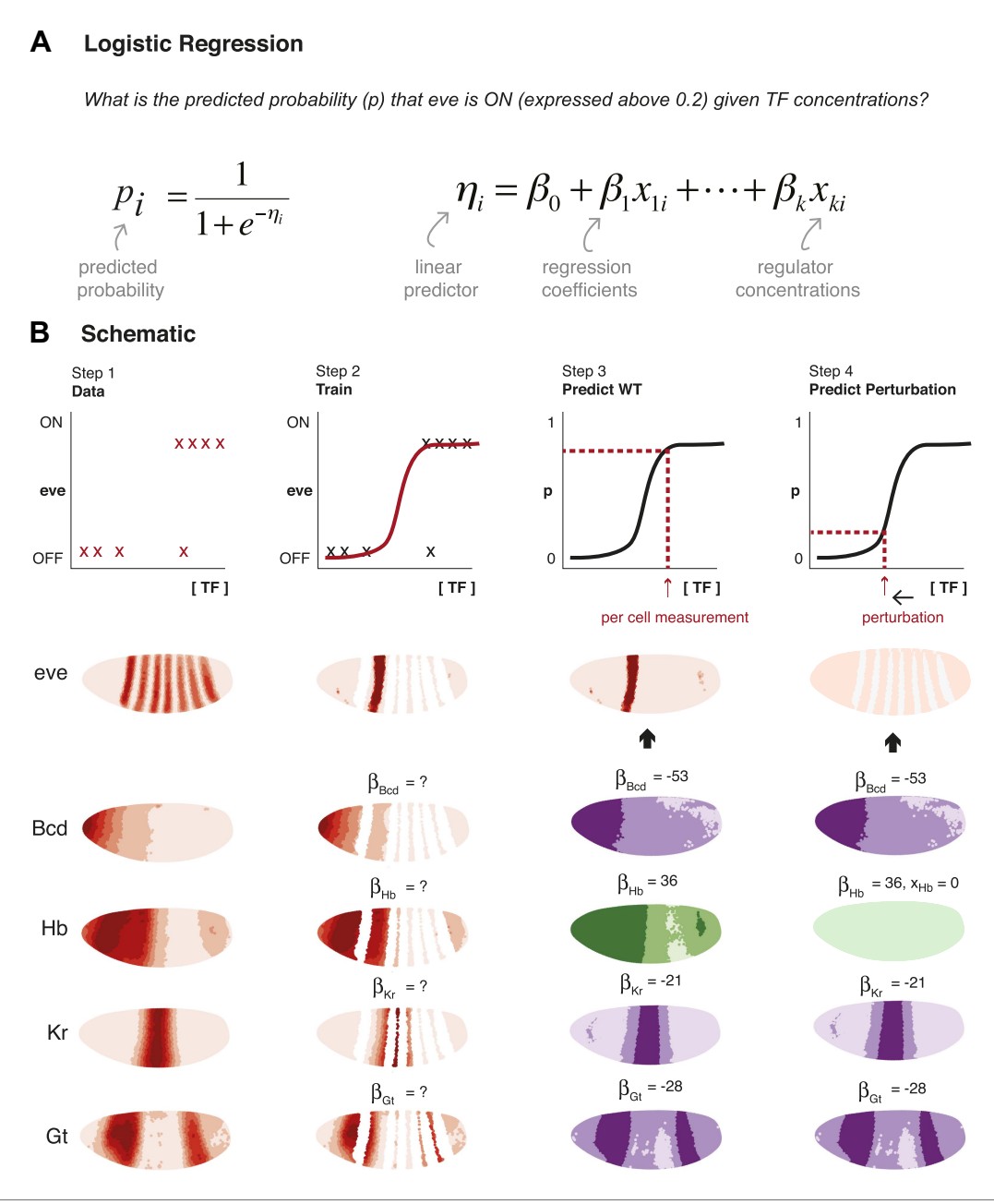

**A  Logistic Regression**

*What is the predicted probability (p) that eve is ON (expressed above 0.2) given TF concentrations?*

$$p_i = \frac{1}{1+e^{-\eta_i}} \qquad\qquad \eta_i = \beta_0 + \beta_1 x_{1i} + \cdots + \beta_k x_{ki}$$

predicted probability — linear predictor — regression coefficients — regulator concentrations

**B  Schematic**

Step 1 Data — Step 2 Train — Step 3 Predict WT — Step 4 Predict Perturbation

per cell measurement — perturbation

$\beta_{Bcd} = ?$ — $\beta_{Bcd} = -53$ — $\beta_{Bcd} = -53$

$\beta_{Hb} = ?$ — $\beta_{Hb} = 36$ — $\beta_{Hb} = 36, x_{Hb} = 0$

$\beta_{Kr} = ?$ — $\beta_{Kr} = -21$ — $\beta_{Kr} = -21$

$\beta_{Gt} = ?$ — $\beta_{Gt} = -28$ — $\beta_{Gt} = -28$

**Figure 1**. Schematic representation of method used to model *eve* expression. (**A**) Logistic regression is used to calculate the probability $p_i$ that *eve* is ON in a given nucleus *i*, given TF concentrations. A logistic model linearly combines the values of independent variables (in this case, the concentrations, $x_{ki}$, of regulators 1 to *k*) to produce a prediction; the predictor, $\eta_i$, is then transformed by the logistic function to give the probability, $p_i$, of *eve* being ON. The weight parameters $\beta_k$ are optimized to provide the best fit with the training data: positive weights indicate activators and negative weights indicate repressors. (**B**) Schematic representation of the data preparation, model training and prediction steps using *eve* stripe 2 as an example. The plots represent how logistic regression operates; the lateral perspectives of the embryo show the Virtual Embryo and processed expression data for *eve* and four regulators (Bcd, Hb, Kr, and Gt). In Step 1, *eve*'s expression is discretized whereas TF concentrations are retained as continuous values. Each nucleus corresponds to a data point. In Step 2, the logistic model is trained to classify whether *eve* is ON or OFF using all nuclei in stripe 2, and all OFF nuclei in the embryo. In Step 3, the trained model is used to predict *eve* expression in every nucleus of the entire embryo using the concentrations of the relevant regulators within them (shown in green
*Figure 1. Continued on next page*

*Figure 1. Continued*

for activators, and purple for repressors). In Step 4, the effects of perturbations are predicted by adjusting the concentration of the regulator under consideration (in this case, Hb), but without changing any model parameters.

The following figure supplements are available for figure 1:

**Figure supplement 1**. Expression of *eve* across the anteroposterior axis.

gastrulation, when according to the Virtual Embryo data, the borders are sharpening and the stripes are not moving dramatically (*Figure 1—figure supplement 1*). This gave us confidence that *eve* is transcriptionally active in the relevant nuclei. At this time point, *eve*'s expression changes from high to low over only a few nuclei across all seven stripes along the anteroposterior axis. Thus we categorized each nucleus as ON or OFF depending on whether *eve* is above or below a value of 0.2 since this defines the stripe borders reasonably (*Figure 1—figure supplement 1*). The Virtual Embryo dataset contains expression measurements normalized to a range of 0 to just over 1 across the entire embryo and time points; thus 0.2 corresponds to ~20% of the maximal expression.

We made use of mRNA measurements for 34 regulatory genes and protein measurements for an additional four genes (*bicoid*, *hunchback*, *Kruppel* and *giant*). Since our model does not require absolute concentration measurements, mRNA expression is a reasonable proxy for protein assuming that the spatial distribution of the two is similar. We distinguish between protein and mRNA measurements by indicating the regulator name in italics for mRNA (e.g., *gt*) or normal case for protein (e.g., Gt). In contrast to *eve*, the expression profiles of these regulators were retained as continuous measurements because many of them are expressed in a graded fashion. Four pair-rule genes (*fushi tarazu*, *odd skipped*, *hairy* and *paired*) that have similar stripe patterns to *eve* were excluded from the data set; although some of them might help modulate the expression of *eve*, they were removed so that we could assess whether *eve*'s complex spatial pattern could be derived directly from simpler patterns of the regulators upstream of the pair-rule genes. Moreover, *eve* expression looks qualitatively normal in these TF mutants (*Schroeder et al., 2011*).

## Modeling *eve* expression using logistic regression

We selected logistic regression for modeling *eve* expression because it provides a framework for linking continuous input variables (i.e., the regulator concentrations) to a binary output (i.e., *eve*'s expression state). Like linear regression, a logistic model linearly combines the values of the independent variables to produce a prediction; but the linear predictor is further transformed by the logistic function to give the probability, $p$, of *eve* being ON (*Figure 1* for a schematic of 'Methods'). As in any regression model, the weight parameters are optimized so that the output shows the greatest agreement with the training data. The weight assigned to each TF indicates its regulatory role, with positive weights indicating activators and negative weights indicating repressors. Importantly, since each regulator in the linear combination has independent weight parameters, the model needs only relative concentration measurements. Models were trained for classification using the nuclei of the stripe(s) under consideration as well as all OFF nuclei in the embryo. It is important to note that although this can be viewed as a training step, the ability of the model to classify at this stage is of direct interest to us: do the regulators contain sufficient positional information to explain *eve* expression in the given stripe(s)? We then use the model to predict *eve* expression in every nucleus across the entire embryo using the concentrations of the relevant regulators within them. This step reveals the model's applicability across the whole embryo, rather than just for the nuclei that were used for training.

## Linear logistic modeling accurately recapitulates *eve 2* expression

First we focused on the expression of the second stripe of *eve*, as it is directed by a very well-characterized enhancer, *eve 2*. Through detailed molecular analysis, it is known that *eve* stripe 2 is controlled by the gap genes (*Frasch and Levine, 1987*; *Stanojevic et al., 1991*; *Small et al., 1992*), a class of TFs that are present in broad regions of the early embryo. In the generally accepted minimal mechanism, two activators Hunchback (Hb) and Bicoid (Bcd) enable broad permissive *eve* expression in the region of stripe 2, while two repressors Giant (Gt) and Kruppel (Kr) define the anterior and posterior borders respectively by suppressing *eve* outside the stripe.

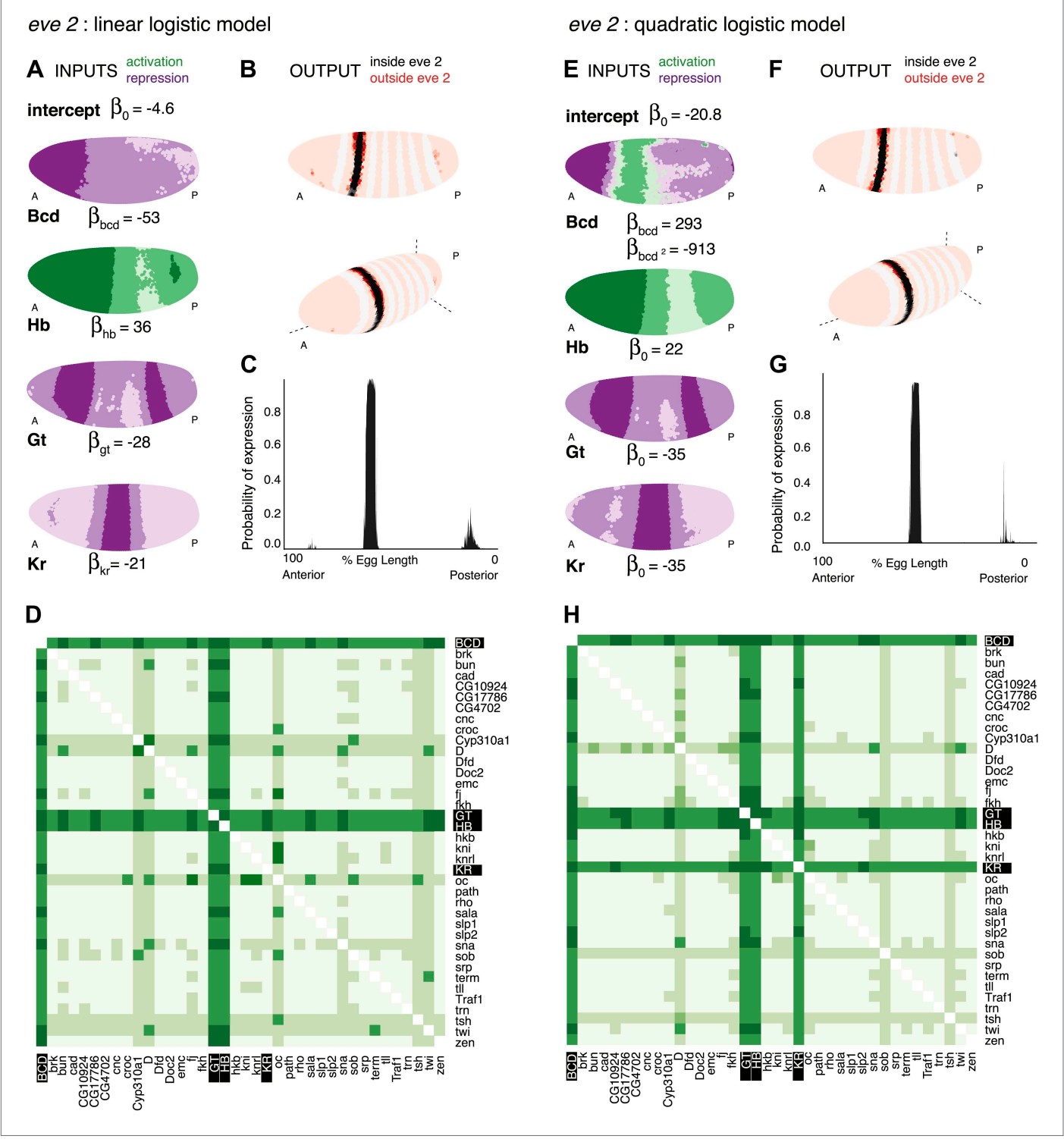

**Figure 2**. Logistic models accurately predict *eve 2* expression. (**A**) Lateral perspectives of the *Drosophila* embryo depicting the contribution of four regulators (Bcd, Hb, Gt, and Kr) to the model output. Embryos are drawn with the anterior (A) to the left and posterior (P) to the right, along with regulator names and corresponding coefficients in the model. Each nucleus is shaded to indicate the level of contribution by regulators, with darker colors signifying stronger effects (in this case, due to higher regulator concentrations): green represents a positive, activating effect and purple a negative, repressive one. Inputs are continuous, but drawn using a discrete color scale for simplicity. (**B**) Lateral and 3D perspectives of the embryo show the model prediction of *eve* stripe 2 expression. Each nucleus is colored from light to dark for low to high probability of *eve* being ON: within stripes the
*Figure 2. Continued on next page*

*Figure 2. Continued*

color scale is from white to black and outside the stripes it is on a red scale, with peach for values below 0.15. (**C**) A ribbon plot showing the probability of *eve* expression (y-axis) for nuclei within 10 μm of the lateral midline along the anteroposterior axis (x-axis). The plot demonstrates that the stripe borders are sharply defined. It also allows easy comparisons with other models that are generally performed in one dimension. (**D**) For regulator discovery, for every possible pair of regulators, we determined the best-scoring model of four regulators containing the pair. The 38 regulators in the dataset are shown on the x- and y-axes of the heat map, and the highest scores for every pair are depicted in the intersecting cell on a color scale from light (minimum score in the heat map) to dark (highest score in the heat map). Regulators making consistently informative contributions to models can be identified by the dark bands running across the heat map. Using linear logistic models, Gt, Hb and Bcd can be clearly seen to be informative regulators (highlighted in black). (**E–G**) Prediction made using a quadratic logistic model, in which Bcd is assigned a concentration-dependent dual regulatory activity: it is an activator (green) at low concentrations in the region of stripe 2, and a repressor (purple) at higher concentrations everywhere else. The model outputs a better prediction for stripe 2 expression as shown in **Table 1**. Most importantly, it reconciles Bcd's apparently paradoxical behavior compared with the literature (**Small et al., 1992**; **Andrioli et al., 2002**). (**H**) Regulator discovery using quadratic models identifies Gt, Hb, Bcd, and Kr as informative regulators (highlighted in black).

The following figure supplements are available for figure 2:

**Figure supplement 1**. Training the linear model without the anteriormost region gives Bcd an activating role.

**Figure supplement 2**. Consistency of the *eve 2* linear and quadratic models (DV, AP and cross-validation).

**Figure supplement 3**. The linear logistic regression model is not unreasonably flexible: a given set of regulators cannot fit any stripe well.

**Figure supplement 4**. The quadratic logistic regression model is not unreasonably flexible: a given set of regulators cannot fit any stripe well.

## Modeling with known regulators recapitulates *eve 2* expression

We trained the logistic model to define the expression of *eve* stripe 2 using a linear combination of the measured concentrations of Hb, Bcd, Gt, and Kr (**Figure 2A**). **Figure 2B** shows the model's output for every nucleus plotted from two perspectives according to their coordinates in the Virtual Embryo. Every nucleus is assigned a probability of *eve* expression and the color scale ranges from light (p=0) to dark (p=1); nuclei within the stripes (defined by actual *eve* expression) are shown in grey-scale from white to black, and predictions outside stripes are presented on a red-scale with peach for values near 0. **Figure 2C** depicts the probability of *eve* expression being above the threshold in the nuclei of the lateral midline along the anteroposterior axis. It is immediately apparent that the model successfully combines the four known regulators to define precisely the location of *eve* stripe 2. (Bcd's role as a potential repressor is discussed below).

To the best of our knowledge, this the first time that *eve 2*'s expression has been predicted so accurately across the entire embryo including the anteriormost region. Most nuclei inside the stripe are correctly classified as having a high probability of being ON, and there is minimal 'over-spill' either side of the stripe (**Figure 2A–C** and **Table 1**). The model defines *eve* expression around the entire circumference of the embryo, following the dorsal-ventral curvature of the stripe: this demonstrates that the four standard regulators of *eve 2* already encode this information, implying that dorsoventral factors are not required to provide this information directly to *eve 2*. Finally, it is

**Table 1.** Measurements quantifying the accuracy of *eve 2* predictions

| Model | In the stripe (%) | Immediate neighbors (%) | 2nd degree neighbors (%) |
|---|---|---|---|
| *eve 2* linear logistic | 86 | 23 | 2 |
| *eve 2* quadratic logistic | 93 | 21 | 2 |

The table shows the percentage of nuclei with predicted *eve* expression (p>0.5). This is a stringent measure of the accuracy of classification, and is particularly useful for assessing the accuracy of the stripe borders. Nuclei with *eve* expression>0.2 are defined as those 'in the stripe'; neighboring nuclei are outside this thresholded region, either immediately adjacent to it, or two nuclei away. A perfect prediction should identify all stripe 2 nuclei as having a high probability of ON with the probability of being ON dropping off rapidly further from the stripe. Though both linear and quadratic models output excellent predictions, the latter provides a slightly more accurate fit to the data.

notable that the model predicts a small amount of expression near stripe 7; ectopic expression of stripe 7 is sometimes observed in transgenic reporters driven by *eve 2* enhancers (*Small et al., 1992*; *Janssens et al., 2006*; *Hare et al., 2008*).

It is worth noting here that the model's performance is most reliably assessed by visually comparing the predicted and actual distributions of *eve* expression as in *Figure 2*; this enables one to evaluate thousands of individual predictions, as well as the overall shape of the prediction, which are not easily captured in a single statistical measure. Nonetheless, *Table 1* quantifies the accuracy of the model— particularly in defining the borders of the stripe—by calculating the percentage of nuclei with a high fitted probability of *eve* being ON (threshold p>0.5; see 'Methods' for description of p). Almost all nuclei within the stripe are correctly identified as ON, and the percentage of nuclei having a high fitted probability quickly drops off further from the stripe.

## The model performs consistently across different subsets of the data

We were interested in the extent to which our model fit is dependent on the subset of the embryo chosen as training data (*Figure 2—figure supplement 2*). We found that the model performs well in a cross-validation test in which we averaged 100 predictions with the training data restricted to 50 randomly selected nuclei; that is, less than 2% of the training data. Conversely, we also assessed whether the model is overly flexible in being able to train and predict the expression of any arbitrary stripe in the embryo using the above four regulators. We found this is not the case, suggesting that the positional information provided by these regulators is specific for the *eve 2* enhancer and that our model interprets this information accurately (*Figure 2—figure supplement 3*).

## Regulator discovery ascertains the known regulators

Having successfully applied the model using known regulators, we next developed a method to identify a parsimonious set of regulators from the dataset informative for the target enhancer's expression. Such techniques are broadly applicable in discovering potential regulators of uncharacterized enhancers, and therefore useful in producing testable hypotheses. We tested a stepwise selection process, but found that it generally includes more regulators than necessary for a good visual fit (e.g., a stepwise selection procedure for *eve 2* with the Bayesian information criterion finds 11 regulators). The stopping point (i.e., the penalty for adding an extra parameter) is effectively arbitrary in this case, or at least difficult to determine a priori in a justifiable manner. Additionally, stepwise selection does not consider all models exhaustively.

Instead, since we are particularly interested in identifying parsimonious models that can explain *eve* expression, and logistic models are fast to fit, we took the approach of fitting all possible models of four regulators out of the possible 38 in the dataset (73,815 models) and used the log likelihood of each fitted model as its score (or equivalently here, the Akaike information criterion). Gratifyingly, the best-scoring model comprises the known regulators Hb, Bcd, Gt, and Kr.

To make use of the scores more generally, we developed a method that summarizes the scores for all 73,815 predictions and highlights regulators that work well together (*Figure 2D*). For each possible pair of regulators (the fixed pair), we determined the best-scoring model of four regulators containing the pair. The two regulators of the fixed pair are shown on the axes on the heat map, with the highest score for the pair depicted in the intersecting cell on a color scale from light (the minimum score on the heat map) to dark (the maximum score). Dark bands crossing the heat map highlight individual regulators that consistently make informative contributions to stripe 2 expression. Hence, it is clear that Bcd, Hb, and Gt are key regulators. Although Kr is actually in the top-scoring model, the heat map does not show it as consistently informative.

## A quadratic logistic model suggests a dual regulatory role for Bcd

The linear model successfully recapitulates stripe 2 expression; however, it identifies Bcd as a repressor, whereas most existing literature defines the TF as an activator. Despite the apparent consensus, Bcd's function is not straightforward. The need for Bcd-binding sites for successful *eve* expression suggests an activating function (*Small et al., 1992*); but this does not explain why the enhancer is inactive in the anteriormost region of the embryo despite Bcd being present at high concentrations and the known repressors Gt and Kr having low concentrations (*Figure 2A*). Our linear models reflect this apparent paradox: Bcd is highlighted as one of the most important TFs during regulator discovery in spite of consistently having a negative coefficient, but a model trained by excluding the anterior region of the embryo assigns Bcd an activating function (*Figure 2—figure supplement 1*).

These observations strongly suggest that Bcd—as both a repressor and activator—provides useful positional information to *eve*.

We asked whether these two functions could be reconciled if Bcd's regulatory effect were dependent on its concentration, either directly, or mediated through other factors or post-translational modifications (*Janody et al., 2000*, *2001*; *Andrioli et al., 2002*). This is readily modeled by adding a single parameter: a quadratic term for Bcd (*Figure 2E–G*). The result is clear: the modified model retains a repressive function for Bcd in the anterior of the embryo where it is present in high concentrations, but enables an activating function in the region of stripe 2 where it has lower concentrations (*Figure 2E*). The modification doesn't lead to over-fitting on small training subsets and in fact improves the model's ability to generalize to the whole embryo from an anteroposteriorly restricted training subset (*Figure 2—figure supplements 2 and 4*). In addition, regulator discovery now identifies all four TFs as important, with a more consistently informative role for Kr than in the simple linear model (*Figure 2H*).

## Independent experiments validate *eve 2* model predictions

We next tested whether our model is predictive of experimental perturbations. We considered experiments that test the role of *eve 2* regulators by either knocking down the input TF (*Stanojevic et al., 1991*), or by mutating binding sites for that TF in the *eve 2* enhancer (*Arnosti et al., 1996*). To simulate these perturbations, we set the concentrations of Bcd or Hb to zero without further

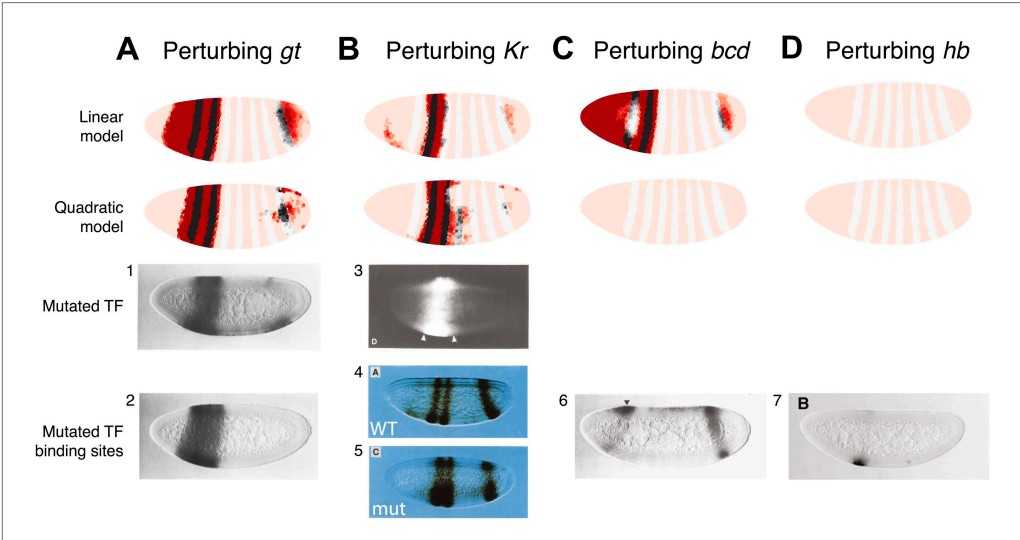

**Figure 3**. The quadratic model accurately predicts *eve 2* expression under perturbation of input TFs. The effects of regulatory perturbations on stripe 2 expression are predicted by altering regulator concentrations but keeping all the model coefficients unchanged; for TF deletion or binding site mutants, this involves setting the relevant regulator's concentrations to 0. Predictions are made for perturbations using the linear and quadratic models. Comparisons to experiments provide robust, independent validations of model predictions. Loss of (**A**) *gt* or (**B**) *Kr* causes *eve* expression to extend towards the anterior and posterior of the embryo respectively, in excellent agreement with experimental evidence. (**C**) For the *bcd* mutant, the linear model predicts expression at the anterior of the embryo, something that is not observed in experiments. In contrast, the quadratic model does not suffer from this. (**D**) Perturbing *hb* leads to complete loss of *eve* stripe 2 for both models. The better agreement between predictions and experimental evidence suggests that the quadratic is a more plausible model of *eve 2* regulation.

In situ images in panels 1, 2, and 6 are reproduced from Figure 4B–C and 6C, **Small et al. (1992)**, *The EMBO Journal*; Nature Publishing Group has granted permission to reproduce these images under the terms of the Creative Commons Attribution 3.0 Unported License (CC BY 3.0). The in situ image in panel 3 is reprinted with permission from Figure 2D, **Small et al. (1991)**, *Genes & Development* (© copyright Cold Spring Harbor Laboratory Press, 1991, All Rights Reserved). In situ images in panels 4 and 5 are reprinted with permission from Figure 3A and 3C, **Stanojevic et al. (1991)**, *Science* (© copyright American Association for the Advancement of Science, 1991, All Rights Reserved). The in situ image in panel 7 is reproduced with permission from Figure 6B, **Arnosti et al. (1996)**, *Development* (© copyright The Company of Biologists, 1996, All Rights Reserved).

adjustment of the coefficients. Strictly speaking, this models the direct effect of the perturbation and is akin to the removal of the relevant binding sites from the enhancer.

The results of these perturbations are shown in *Figure 3*. Only the quadratic model correctly predicts the expression pattern in a Bcd null mutant (*Figure 3C*). In the linear model, Bcd is designated a repressor and so its mutant causes broad *eve* expression in the anterior of the embryo in contrast to the experimental result (*Figure 3C*). In the quadratic model the lack of either activator (Bcd or Hb) abolishes the expression of stripe 2 as expected (*Figure 3C,D*). In both the linear and quadratic models, the loss of the repressors Gt or Kr causes *eve* expression to extend towards the anterior and posterior of the embryo respectively, in line with their roles in defining the stripe borders (*Figure 3A,B*).

Both models predict the observed response to binding site mutations: the expansions of stripe 2 in the correct directions and extent along the length of the embryo. The models demonstrate that the extent of posterior extension in the Kr mutant is restricted because of decreasing activator concentrations (*Figure 3B*). For the anterior extension in the Gt mutant, the restriction requires a repressor since activator concentrations remain high to the end of the embryo (*Figure 3A*; *Andrioli et al., 2002*). Bcd can provide this repression in both linear and quadratic models: however, only the quadratic can reconcile this with Bcd's known activating function. For the linear model to work with a Bcd activator, one would need a fifth regulator as an anterior repressor. Indeed, multiple studies have searched for a repressor in this region, and multiple candidates have been identified though none have been conclusive (*Bellaïche et al., 1996*; *Janody et al., 2000*; *Andrioli et al., 2002*; *Zhao et al., 2002*; *Singh et al., 2005*).

## Models successfully predict *eve* 3+7 expression

*eve* stripes 3 and 7 are regulated together by a single enhancer (*Small et al., 1996*; *Clyde et al., 2003*; *Struffi et al., 2011*). Such an arrangement requires appropriate TF concentrations for *eve* activation to be present in nuclei separated by some distance. We tested whether our modeling framework can contend with the challenge of specifying two extra stripe borders using the available regulator concentrations.

### A combination of modeling and regulator discovery suggests two plausible models

We first fit our models using only the known regulators of *eve* stripes 3 and 7, Hb and Kni. Kni is thought to repress the region between the stripes and Hb is thought to repress in the anterior and posterior regions outside the stripes (*Clyde et al., 2003*). The measured concentrations of Hb (protein) and *kni* (mRNA) alone are not sufficient for our models of stripe 3 and 7 expression; in particular, the concentration of Hb is too low to repress expression to the posterior of stripe 7 (*Figure 4—figure supplement 1*).

Using regulator discovery (*Figure 4—figure supplements 6 and 7*), we identified two alternative models that are able to define stripes 3 and 7 (*Figure 4*). The first is a linear logistic model that includes two additional gap genes, Giant (Gt) and *tailless* (*tll*); including both Gt (protein) and *tll* (mRNA) improves predictions over including *tll* alone (*Figure 4—figure supplement 2*). In this model, all regulators function as repressors: the model has a positive intercept which can represent a ubiquitous activator (*Figure 4A*). Our second model is a quadratic logistic regression model that treats Hb as a dual regulator, in a similar manner to Bcd for *eve 2* (*Figure 4D*). Concentration-dependent regulation by Hb—as an activator at low concentrations and repressor at higher levels—has been suggested by previous experimental work (*Hülskamp et al., 1990*, *1994*; *Zuo et al., 1991*; *Schulz and Tautz, 1994*); and used to model stripe 3 expression (*Papatsenko and Levine, 2008*) and gap gene regulation (*Bieler et al., 2011*). Using regulator discovery, we again identified *tll* as the top candidate for repressing expression posterior to stripe 7 (*Figure 4—figure supplement 7*). *tll* has previously been proposed as a regulator of stripe 7, in some cases as an activator (*Small et al., 1996*) and in others as a repressor (*Janssens et al., 2006*; *Morán and Jiménez, 2006*). Both our linear and quadratic models output good predictions of *eve* stripes 3 and 7 (*Figure 4B,C,E,F* and *Table 2*). As with predictions for *eve 2*, the high probability predictions are within the stripes and the models successfully replicate *eve* expression around the embryo. Further, using these chosen regulators, the models are not able to train and predict the expression of any arbitrary pair of stripes (*Figure 4—figure supplements 4,5*).

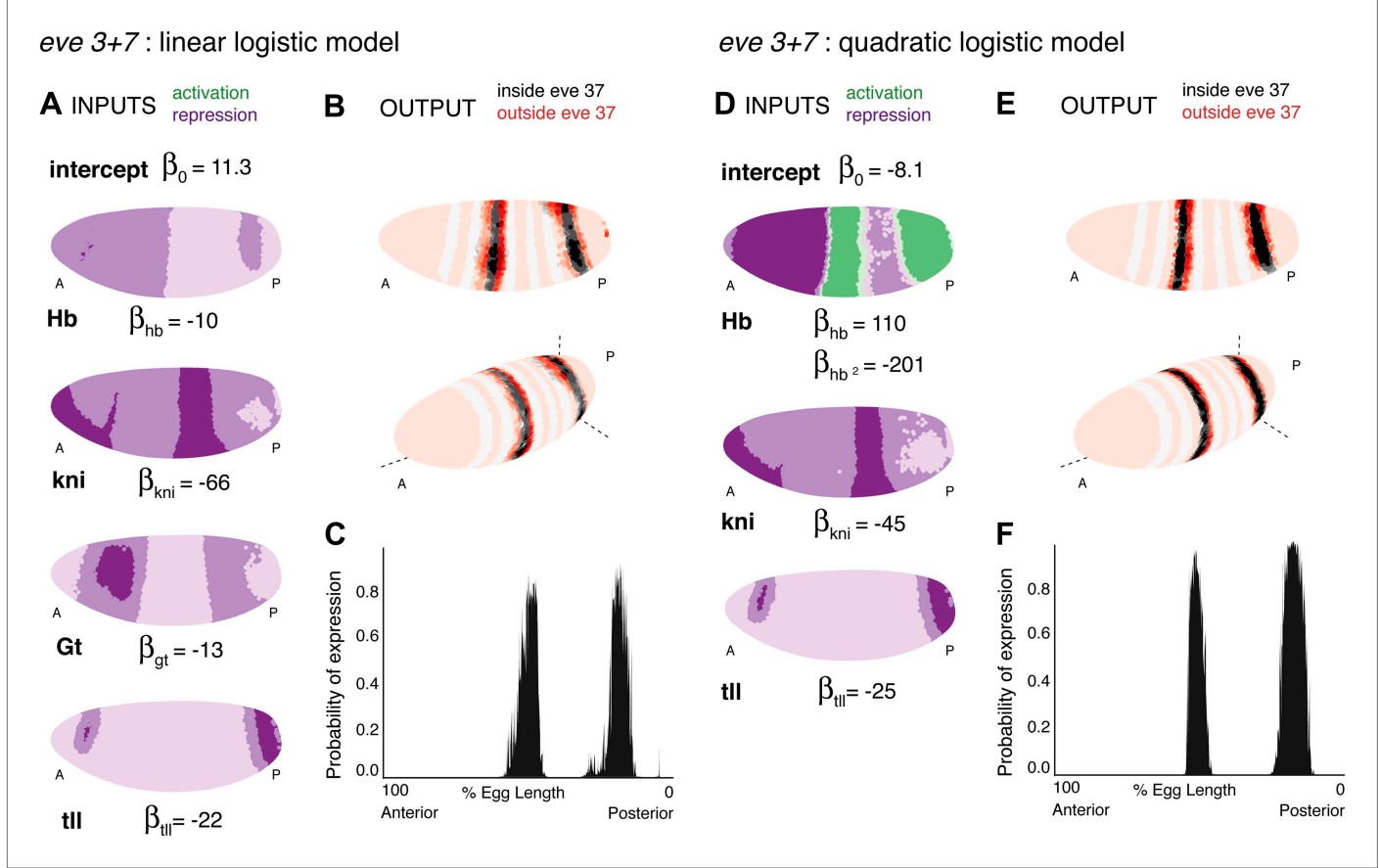

**Figure 4**. Linear and quadratic logistic models accurately predict *eve 3+7* expression. Regulator inputs and model output are shown as in *Figure 2*. (A–C) The linear model including Hb, *kni, tll,* and Gt; (D–F) the quadratic model comprises Hb, *kni,* and *tll,* with a quadratic term for Hb as a concentration-dependent dual regulator. Both models clearly define the two stripes, though the midline ribbon plots show that the quadratic model defines the sharpest borders. The initial predictions therefore suggest that the quadratic model provides the best output.

The following figure supplements are available for figure 4:

**Figure supplement 1**. Hb and *kni* are not sufficient for a good model fit.

**Figure supplement 2**. A linear logistic model with Hb, *kni* and *tll* does not have sharp stripe borders.

**Figure supplement 3**. Consistency of the *eve 3+7* linear and quadratic models.

**Figure supplement 4**. The linear logistic regression model is not unreasonably flexible: a given set of regulators cannot fit any pair of stripes well.

**Figure supplement 5**. The quadratic logistic regression model is not unreasonably flexible: a given set of regulators cannot fit any pair of stripes well.

**Figure supplement 6**. Regulatory discovery for a linear logistic model of *eve 3+7*.

**Figure supplement 7**. Regulatory discovery for a quadratic logistic model of *eve 3+7*.

## There are reasons to favor the quadratic model

We prefer the quadratic model over the linear for a variety of reasons. First, it is simpler: the quadratic requires only three regulators, compared to four in the linear model. Both models have five parameters, which include three shared regulators (Hb, *kni, tll*) and the intercept. Second, the quadratic model has more clearly defined stripe borders than the linear model. Third, the quadratic model is more robust to the choice of training data (*Figure 4—figure supplement 3*), indicating that it describes

**Table 2.** Measurements quantifying the accuracy of *eve 3+7* predictions

| Model | In the stripe (%) | Immediate neighbors (%) | 2nd degree neighbors (%) |
|---|---|---|---|
| *eve 3+7* linear logistic | | | |
| *kni* and Hb | 0 | 0 | 0 |
| *kni,* Hb, and *tll* | 56 | 39 | 27 |
| *kni,* Hb, *tll*, and *gt* | 76 | 37 | 17 |
| *eve 3+7* quadratic logistic | 86 | 39 | 9 |

The table shows similar measures of accuracy for stripes 3 and 7 as in **Table 1**. It is clear that the quadratic model and the 4-regulator linear model provide the best predictions, with the most sharply defined borders.

the regulatory relationship uniformly across the embryo: the model performs consistently whether it is trained on either of the two stripes, a restricted region around the lateral midline, or only on the stripes and their immediately neighboring nuclei. Finally, the quadratic model retains accurate expression of the stripes even when all 38 candidate regulators are included; by contrast the prediction from the linear model begins to fragment spatially, which suggests localized over-fitting.

## Independent experimental perturbations are consistent with the quadratic model

As with *eve 2*, we can further compare the models by predicting the outcomes of regulatory perturbations of input TFs (**Figure 5**). Here we consider perturbations of *kni* and *hb*, the best characterized regulators of *eve 3+7*. It is again important to distinguish between expression in a mutant background, which reveals both direct and indirect interactions, and corresponding binding site mutations within the *eve 3+7* enhancer, which probe only direct interactions.

### Perturbing *kni*

In the *kni* mutant, expression of an *eve 3+7* reporter transgene extends fully between the two stripes before partially retreating towards wild-type expression (**Small et al., 1996**). Similarly, when Kni-binding sites in the *eve 3+7* enhancer are removed, the expression pattern matches the *kni* null mutant (**Struffi et al., 2011**), although an earlier transgenic reporter with fewer mutated Kni-binding sites showed only partial extension (**Clyde et al., 2003**).

To mimic both of these types of perturbations, we eliminated *kni* as an input. Under these conditions, the linear model predicts the observed full extension between the two stripes, whereas the quadratic does not (**Figure 5A**). However, given our reasons for preferring the quadratic model described above, it is worth considering some minor and biologically plausible assumptions to reconcile these perturbation experiments with the quadratic model (**Figure 5C,D**). We discuss these in terms of direct and indirect effects below.

### Direct effects

The direct effects can be understood by considering two related minor adjustments (**Figure 5C**). First, we can assume that the Kni protein is ubiquitously expressed in the embryo at low concentrations, but that this is not reflected in the Virtual Embryo dataset; it is possible that in situ hybridization was not sufficiently sensitive for these low-level transcripts or that the protein has a slightly different profile to the *kni* mRNA. If we increase *kni* concentrations in the Virtual Embryo dataset by just 0.1 (~10% of the maximum measured value across all time points), the retrained quadratic model predicts full extension between the stripes. The prediction for wild-type expression is not affected and this adjustment is sufficient for explaining both the *kni* mutant and the *eve 3+7* binding site mutations. Second, since this adjustment produces a model with identical coefficients (i.e., $\beta_k$) except for the intercept (i.e., $\beta_0$)—which increases from −8.1 to −3.6—we can change the intercept directly in the quadratic model. This adjustment is also sufficient to predict full extension between the stripes, though it alters the wild-type prediction slightly (**Figure 5—figure supplement 2**). This change in the intercept corresponds to potential differences in expression between the endogenous gene and the transgenic reporter. For instance, the reporter might have a lower barrier to activation and be more efficiently transcribed relative to the endogenous enhancer. Alternatively, the mutations in the transgenic enhancer may have abolished Kni repression, but then introduced the binding of another weak, ubiquitous activator.

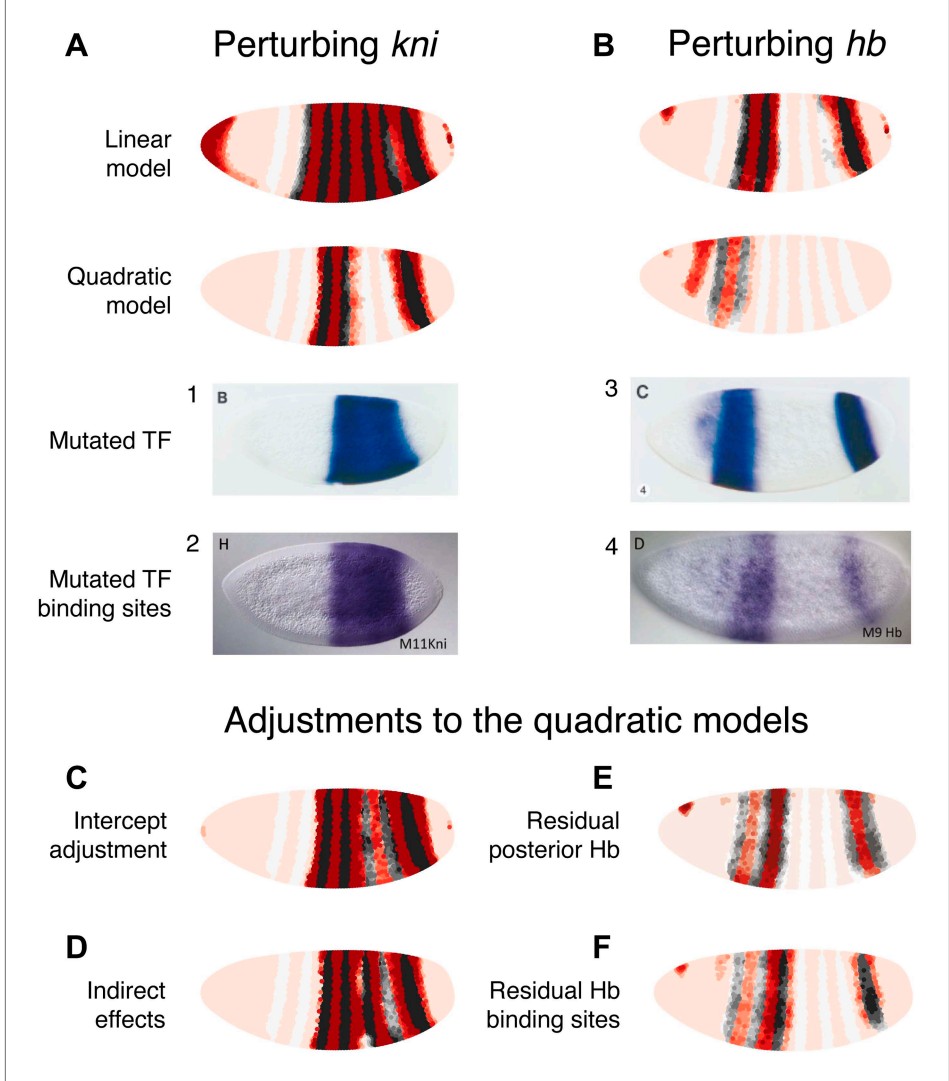

**Figure 5**. Linear and quadratic logistic models accurately predict *eve 3+7* expression under perturbation of Kni and Hb. The effects of regulatory perturbations on *eve 3+7* expression are predicted as described in the main text. (**A**) Perturbation of *kni* and its binding sites cause full reporter expression between the stripes. The linear model predicts this observed extension, but the quadratic does not. (**B**) Perturbation of *hb* causes stripe 3 to expand and move anteriorly, and stripe 7 to expand slightly. Binding site mutations show similar effects, though perhaps without the anterior shift of stripe 3. The linear model provides good prediction of both stripes. The quadratic produces a good stripe 3 prediction, including its anterior shift, but fails to predict any expression in stripe 7. (**C–F**) Given the initial preference for the quadratic, we considered minor and biological plausible assumptions that allow the model to make accurate predictions. For the *kni* mutants, these are (**C**) the minor adjustment of the intercept and (**D**) inclusion of indirect effects of *kni* on *hb* by increasing Hb by 50% of wild-type *kni*. For the *hb* mutants, these are (**E**) the inclusion of residual maternal Hb in the posterior and (**F**) simulating the effects of residual Hb binding sites.

In situ images in panels 1 and 3 are reprinted with permission from Figure 4B–C, *Small et al. (1996)*, *Developmental Biology* (© copyright Elsevier, 1996, All Rights Reserved). In situ images in panels 2 and 4 are reproduced with permission from Figures 4H and 6D, *Struffi et al. (2011)*, *Development* (© copyright The Company of Biologists, 2011, All Rights Reserved).

The following figure supplements are available for figure 5:

**Figure supplement 1**. Indirect effects and the quadratic model can explain the expansion and retreat of expression observed in the *eve 3+7* reporter in a *kni* mutant.

*Figure 5. Continued on next page*

*Figure 5. Continued*

**Figure supplement 2**. An adjustment to the intercept in the quadratic logistic model for *eve 3+7* results in a slight expansion of expression between the stripes.

**Figure supplement 3**. Hb binding site mutants may dampen or remove Hb repression at higher concentrations.

## Indirect effects

Next, we consider the effects of Kni on downstream regulators (*Figure 5D*). There is strong evidence that *kni* is a repressor of *hb*, as its loss causes *hb* expression to extend from stripe 7 towards stripe 3 (*Hülskamp et al., 1990*; *Clyde et al., 2003*). We simulated this indirect interaction by increasing Hb concentration in proportion to the relative loss of *kni*. Since Hb is an activator at low concentrations in the quadratic model, this indirect effect can drive *eve* expression between the stripes. This adjustment is not relevant to binding site mutants, but interestingly it does provide a tentative explanation for the partial retreat of *eve*'s extension towards a wild-type expression pattern: as Hb concentrations increase over time, the TF eventually switches from an activator to a repressor of *eve* between the stripes (*Figure 5—figure supplement 1*).

## Perturbing *hb*

*hb* is both maternally and zygotically expressed. In embryos null for *hb* zygotic expression, *eve* stripe 3 moves anteriorly and expands, whereas stripe 7 shows more limited widening (*Small et al., 1996*). Mutating Hb-binding sites in the *eve 3+7* enhancer leads to similar expansion, though perhaps without the anterior shift of stripe 3 (*Struffi et al., 2011*). Maternally deposited *hb* mRNA is ubiquitous but differentially translated in the anterior (*Hülskamp et al., 1989*). Zygotically, *hb* is transcribed in both an anterior domain that largely overlaps the maternal *hb* pattern and in a posterior stripe (*Margolis et al., 1995*). Thus at the time point used here, the zygotic mutant likely contains residual Hb protein in the anterior at the time point we use in this study.

We simulated the zygotic mutant by eliminating Hb altogether in the posterior and decreasing its expression to 20% in the anterior domain (*Figure 5B*). In these conditions, the linear model produces a good prediction for both stripes. The quadratic model arguably produces a better prediction for stripe 3, capturing its movement towards the anterior; however, it predicts zero expression in stripe 7. Based on our preference for the quadratic model as described in previous sections, and on the evidence presented in the following section, we again consider minor adjustments to reconcile it with experimental results (*Figure 5E,F*).

## Direct effects

The first adjustment is relevant for the zygotic mutant, and assumes some active Hb in the posterior around stripe 7 (*Figure 5E*). Specifically, having as little as 0.15 of Hb (~15% of maximal expression) in this region is sufficient for a good prediction of stripe 7. Next, we simulated the effects of having some residual Hb-binding sites in the enhancer, as a mutagenesis experiment may not abolish all binding (e.g., see *Clyde et al., 2003* compared to *Struffi et al., 2011*). A simple way to model this is to have the same low level of Hb activity as in the zygotic mutant, which produces a good prediction as just described. As an alternative, we also considered whether incomplete mutagenesis could affect Hb's dual-regulatory behavior. *Papatsenko and Levine (2008)* proposed that the dual role is facilitated by adjacently bound Hb molecules masking each other's active sites; in this scenario, we would expect dual regulation to be attenuated as binding sites are lost through mutagenesis. We simulated this by dampening the regulatory effect of Hb at higher concentrations and found that the predicted expression patterns agree with experimental results (*Figure 5F*, *Figure 5—figure supplement 3*). Notably, this prediction does not show movement of stripe 3 towards the anterior or expansion in stripe 7, correctly reflecting the experimental results of binding site mutagenesis.

## Models predict *eve 2* and *3+7* expression at earlier time points

We also tested the linear and quadratic models for both *eve 2* and *eve 3+7* on the two previous time points in the Virtual Embryo, which were not used for training (*Figure 6*). The results for *eve 2* show a wider stripe forming before it narrows to the boundaries of stripe 2. This mirrors published results for an *eve 2* reporter as well as the endogenous expression of *eve* (*Small et al., 1992*; *Andrioli et al., 2002*). The predictions for *eve 3+7* are also consistent in terms of the positions of the stripes, although

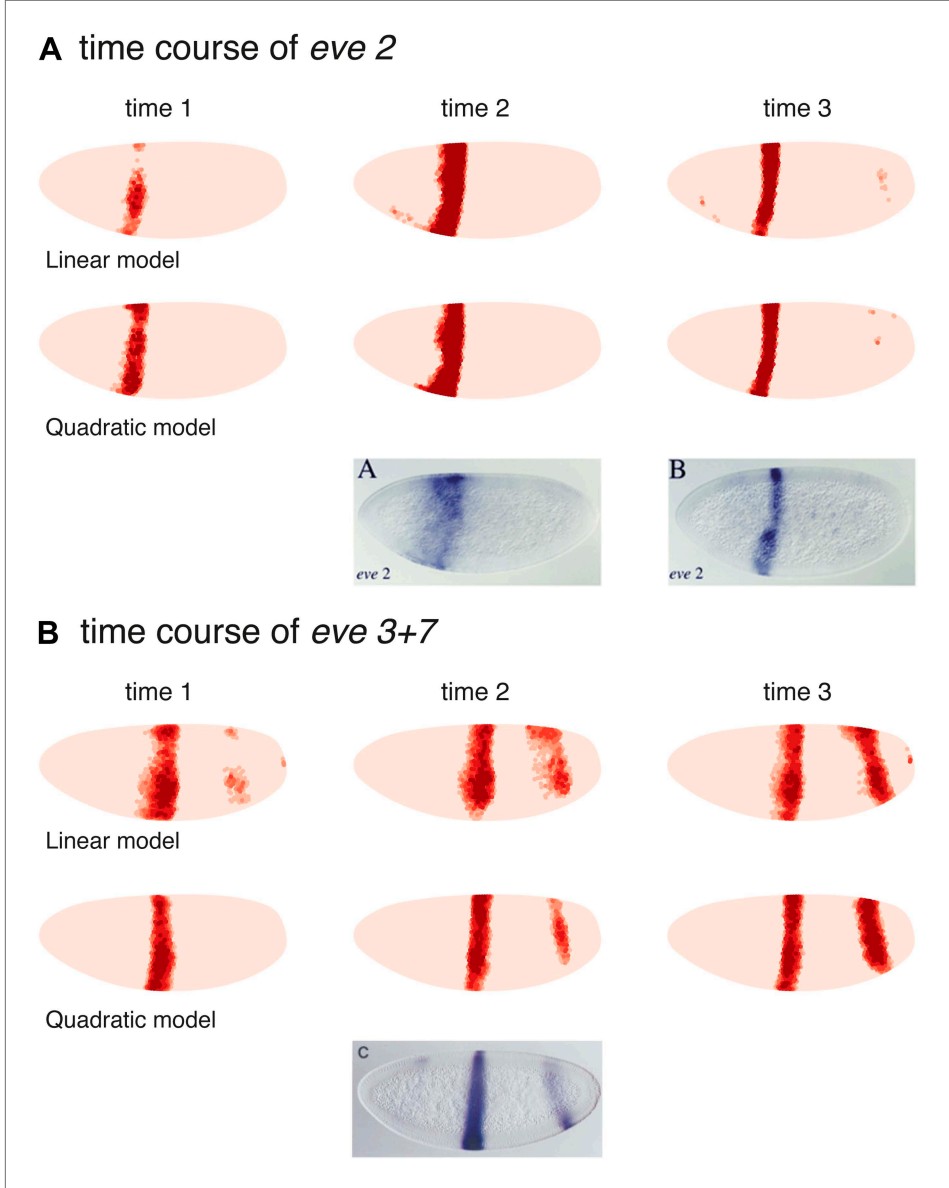

**Figure 6**. Models predict *eve 2* and *3+7* expression at earlier time points. Model predictions for earlier time points in the Virtual Embryo are shown for the (**A**) *eve 2* and (**B**) *eve 3+7* linear and quadratic models. The time points are labeled from the start of the dataset; the third time point is the one used throughout the main text. For *eve 2*, the linear and quadratic models show a wider stripe at the second time point and a well-defined stripe at time point 3. This matches the in situ images below from ***Andrioli et al. (2002)*** which show a transgenic reporter at early and mid cycle. The predictions for *eve 3+7* are consistent in terms of the positions of the stripes, with stripe 3 appearing earlier than stripe 7. At the earlier time points the difference in sharpness of the stripe borders between the quadratic and linear model is more pronounced suggesting that the interpretation of positional information by the quadratic model is more stable and precise.

The in situ image for *eve 3+7* is reprinted with permission from Figure 2C, ***Small et al. (1996)***, *Developmental Biology* (© copyright Elsevier, 1996, All Rights Reserved). In situ images for *eve 2* are reproduced with permission from Figure 4A,B, ***Andrioli et al. (2006)***, *Development* (© copyright The Company of Biologists, 2006, All Rights Reserved).

they have stripe 3 appearing earlier than stripe 7. This timing difference is not obvious in the endogenous *eve* expression recorded in the Virtual Embryo, although stripe 7 does appear relatively weak in some transgenic reporters (***Small et al., 1996***). At the earlier time points the difference in sharpness of the

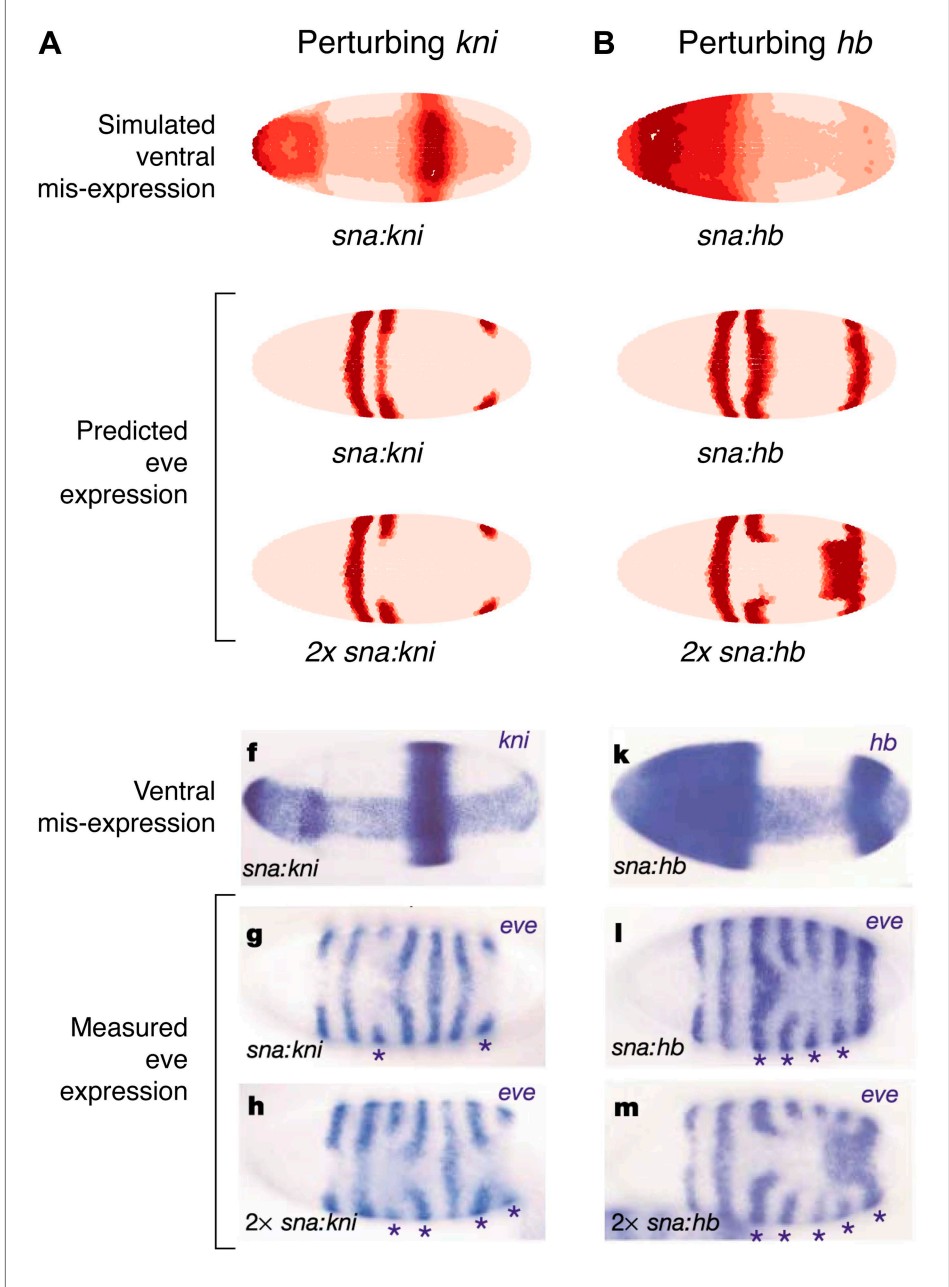

**Figure 7**. Quadratic models accurately predict fine-scale features of expression patterns due to input misexpression. The study by **Clyde et al. (2003)** misexpressed *hb* and *kni* along the ventral surface of the embryo using transgenes driven by a *snail* promoter and recorded the effects of one or two copies of these transgenes on *eve* expression. We replicated these experiments using quadratic models for *eve 2* and *eve 3+7* (trained on stripe 3), by adding Hb and *kni* in proportion to the distribution of *snail* in the Virtual Embryo dataset. As described in the main text, we also added an indirect effect from Hb activating *Kr*. (**A**) With *kni* misexpression, the model accurately predicts the thinning (x1 transgene), then cutting of stripe 3 (x2). (**B**) With Hb misexpression, the model successfully predicts the bulging, then cutting and bending of stripe 3 (x2), and the bulging of stripe 7 (x2). Stripe 2 remains unaffected in both perturbations, in agreement with the experimental results. The accuracy of the predictions indicates that the quadratic model for *eve 3+7* can explain the experimental results very well. In contrast the linear models are unable to predict these results.

In situ images are reproduced from Figures 1F–H and 1K–M, **Clyde et al. (2003)**, published in *Nature*; Nature Publishing Group has granted permission to reproduce these images under the terms of the Creative Commons Attribution 3.0 Unported License (CC BY 3.0).
*Figure 7. Continued on next page*

*Figure 7. Continued*

The following figure supplements are available for figure 7:

**Figure supplement 1**. Supplementary misexpression predictions of the *eve 2* and *eve 3+7* linear (A,B) and quadratic (C–E) models.

stripe borders between the quadratic and linear model is more pronounced suggesting that the interpretation of positional information by the quadratic model is more stable and precise.

## Quadratic *eve 2* and *3+7* models predict TF misexpression results better than linear models

As a final test of our models, we compare our predictions to the misexpression work of Clyde and colleagues (*Figure 7*, *Figure 7—figure supplement 1*; *Clyde et al., 2003*). In this study, the authors constructed transgenes with a *snail* promoter that misexpressed *hb* or *kni* in the ventral region of the embryo and recorded the effects of expressing one or two copies of these transgenes. The experiments confirmed that Hb and Kni repress *eve 3+7* and *4+6* at different concentrations as suggested by *Fujioka et al. (1999)*. However, the experiments also revealed some curious observations that are not easily explained. First, stripes 3 and 7 respond differently to the same additional concentrations of Hb despite being regulated by the same enhancer. Secondly, and most intriguingly, the results show substantial bending of stripes: in the presence of one copy of the *hb* transgene, stripe 3 extends towards the posterior of the embryo, and with two copies, stripe 7 bends towards the anterior. These behaviors cannot be explained readily by simple, qualitative inspection of the embryos and they were not explored in the original study.

To model this experiment, we added Hb and *kni* to the Virtual Embryo in proportion to the measured distribution of *snail* at different concentrations ranging from 0.1 to 0.4 (~10% to ~40% of maximal expression). We simulated the responses of *eve 2* and *eve 3+7* using both the linear and quadratic models above and a quadratic model trained on stripe 3. *Figure 7* and *Figure 7—figure supplement 1* display the predictions of these models at 0.2 and 0.4 of Hb and *kni*, simulating the effects of one or two copies of the transgene respectively. *Figure 7* includes a putative indirect effect on *eve 2* via Kr (see below).

The quadratic models predict the fine-scale effects of this misexpression experiment with remarkable accuracy whereas the linear models do not (compare *Figure 7* to *Figure 7—figure supplement 1A,B*). Specifically, the quadratic model trained on stripe 3 successfully reproduces the bending and bulging of both stripes 3 and 7 under *hb* misexpression (*Figure 7B*), as well as the repression seen with *kni* misexpression (*Figure 7A*). Our models for *eve 2* predict that since Hb is an activator, increasing the concentration of Hb should lead to increased activation of *eve 2* and a resultant broadening of the stripe (*Figure 7—figure supplement 1A,C,E*); but this is not in fact observed in the misexpression study. Indirect effects of Hb on another factor, such as Kr, can resolve this discrepancy: adding Kr in proportion (50%) to the increase of Hb does indeed prevent stripe 2 from expanding, but only in the quadratic model and not in the linear (*Figure 7* and *Figure 7—figure supplement 1B,D*). Given all the evidence provided here, on balance we conclude that the quadratic models are the most likely for both *eve 2* and *eve 3+7*.

## Discussion

### Summary

Our goal was to understand the regulatory system underlying spatiotemporal patterning in the early *Drosophila* embryo by fitting regulatory input functions to the output of individual enhancers. Our models are accurate, predictive and simple to apply and interpret. We showed that simple functional forms relating TF concentrations to *eve* expression outputs are highly predictive of wild-type and mutant expression patterns. In doing so, we have demonstrated that precise positional information—in other words, information interpreted by individual nuclei to produce an expression pattern—is available in the early embryo. We determined whether TFs are most informative when serving as activators or repressors of each enhancer, and we also explored whether a dual-regulatory role for some TFs improved expression predictions. Here we discuss our work in relation to other models of regulatory function, the insights our models provide into transcriptional regulation and positional information in the embryo, and the experimentally testable hypotheses proposed by our models.

## Previous models

The regulation underlying anteroposterior patterning of the *Drosophila* blastoderm has long been a favorite system for modeling work; for recent reviews, see for example *Jaeger et al. (2009)* and *Papatsenko (2009)*. Some models have been successful in reproducing the gap gene patterns (*Jaeger et al., 2004*, *2007*; *Bieler et al., 2011*; *Papatsenko and Levine, 2011*), but none have succeeded in accurately predicting precise stripes of *even skipped* across the whole anteroposterior axis of the embryo (*Levine, 2008*). In general, previous models have focused on utilizing information contained in the *cis*-regulatory sequence; for example predicting expression and evaluating potential TFs of a 1.7kb region of regulatory DNA upstream of *eve* (*Janssens et al., 2006*), fitting models to fusions of *eve* enhancers and predicting expression from different regulatory DNA (*Kim et al., 2013*), testing models of TF binding and synergy by predicting expression across many enhancers (*Segal et al., 2008*; *He et al., 2010*) and identifying enhancer sequences within the genome based on the fit between predicted and observed expression patterns (*Kazemian et al., 2010*).

Our choices in modeling the regulatory function of enhancers differ from these previous studies in a number of important respects. First, our models are highly accurate in fitting the *eve* expression pattern in the entire embryo. This is in part because we chose to model the regulatory function of each enhancer separately, rather than fitting a single model that applies across many enhancers simultaneously. By defining parameters that are specific for each enhancer, we are able to assign the regulatory roles for TFs in a context-specific manner. Second, our models also perform well because, unlike previous studies, we do not impose any biological mechanisms on our models (e.g., a 'thermodynamic score' for protein–DNA interactions). Instead we worked the other way round: we tested models that fit data as accurately as possible and then inferred the underlying mechanisms. This simple framework nonetheless allows us to propose experimentally testable hypotheses. Third, our modeling framework is quick to apply. This allowed us to search comprehensively for informative regulators, a property that is particularly valuable for studying poorly characterized enhancers.

## Inferred mechanisms for regulatory input function

Since the models are accurate and predictive, they may reflect the underlying molecular mechanism for transcriptional regulation. Further, the models are relatively easy to interpret, so we can infer what they mean in terms of biological mechanism. Here we highlight three features.

### Thresholding a combination of TFs is sufficient for positional information

One of the important questions in animal development is how each cell determines its position in the embryo. Early work on positional information in the *Drosophila* embryo was inspired by the idea of a morphogen gradient that is interpreted by the nuclei according to a set threshold (*Ashe and Briscoe, 2006*; *Crick, 1970*; *Wolpert, 1969*, *1996*). More recently, it has been concluded that a lone-acting morphogen is insufficient for providing precise positional information to the embryo, especially since no gradient with this characteristic has been measured in an embryo (*Kerszberg and Wolpert, 2007*; *Wolpert, 2011*). Our model, however, shows that the combined action of multiple morphogens and a corresponding interpretative threshold is indeed able to read positional information from measured gradients alone. In particular, it succeeds by applying the threshold to the overall balance of activators and repressors rather than to each factor individually.

Focusing on the contributions of individual TFs also tends to emphasize the role of repressors in providing positional information to the *eve* stripes. Since repressors are often crucial in defining the borders of the stripes, it is natural to suppose that the activators are merely permissive, and that precision in positional information is provided by the repressors. Here we show that an alternative view is compatible with the data. In particular, activators and repressors contribute symmetrically to positional information: they work to increase or decrease the probability of transcription, but neither class acts separately according to a threshold that is independent of the concentrations of other factors. Thus, if more activators are present in a nucleus, a higher concentration of repressors will be required to reduce transcription to the same level. This means that positional information cannot be defined by any one factor in isolation, and nor can mutant results be interpreted reliably in the absence of data on other factors.

### Pairwise cooperative interactions between TFs are not necessary for synergy

Our model can help clarify the concept of synergy, where the effect of one regulator depends on the concentration of another. This has been proposed in the context of transcriptional activators in general

(*Struhl, 2001*) and observed between Hb and Bcd in controlling expression of *eve 2* (*Stanojevic et al., 1991*; *Simpson-Brose et al., 1994*). Our model shows that this effect is observed with a linear combination of concentrations: that is, without any pairwise interactions between Hb and Bcd or other factors. Thus, our model is compatible with the early findings of *Arnosti et al. (1996)*, which suggest that *eve* transcription is controlled by the total balance of activators and repressors rather than through complex and intricate combinatorial interactions between TFs. However, this is not to say that cooperative interactions do not take place, or are not important in other contexts, but rather that it is necessary to distinguish between synergistic interactions that can be explained by independent binding of multiple factors (as in our model), and those that occur as a result of pairwise interactions between TFs. We expect pairwise interactions between TFs on the DNA to require particular arrangements of binding sites. Therefore, the success of our model without pairwise interactions suggests that the ordering and exact spacing of binding sites are not important, except potentially in the case of dual regulation. This agrees well with multiple observations about the flexibility of enhancer sequences, which can tolerate rearrangement over evolutionary time while maintaining their function (reviewed in *Borok et al., 2010*).

## Dual regulatory function of Hb and Bcd

Hb has a dual role: it acts as a repressor in some enhancers (e.g., *eve 3+7*) and an activator in others (e.g. *eve 2*) (*Small et al., 1992*, *1996*). Here, like *Papatsenko and Levine (2008)*, we model a dual role for Hb in the context of a single enhancer. In our model of *eve 3+7*, including a quadratic term for concentration-dependent dual regulation produces better wild-type predictions, explains experimental perturbations accurately (with certain assumptions), and produces consistent fits across different training subsets. Although *Kazemian el al. (2010)* did not find a quadratic term for Hb generally useful for fitting logistic models to *Drosophila* expression patterns, they did find this to be true for Bcd, particularly for the anterior parts of the expression patterns. In our work, this term is not needed for a good fit, but we add it for *eve 2* to show how a repressive role in the anterior can be reconciled with an activating role around stripe 2 (*Zhao et al., 2002*; *Singh et al., 2005*).

Concentration-dependent regulatory activities have been observed in other systems: for instance in humans, at low concentrations, Sp1 is as an activator of the folate receptor gene in conjunction with Ets TFs; at higher concentrations it becomes a repressor by blocking Ets binding (*Kelley et al., 2003*). Our model does not reveal how Hb and Bcd achieve dual-regulatory activity, and it is quite possible that they make use of different mechanisms. One possibility is a change in protein–protein interactions, through formation of homo-oligomers or interactions with co-factors (e.g., *Janody et al., 2001*). Alternatively as with Sp1, changes in DNA occupancies may alter how regulators interact with adjacent TF molecules. We discuss experimental tests of these possibilities below. Regardless of mechanism, however, we propose that concentration-dependent effects are important, in contrast to the hypothesis that concentrations above a predefined threshold are neutral in effect. Moreover, we suggest that similar analysis techniques could be used to test potential dual-regulatory capabilities of other regulators, such as Gli and Lef/Tcf in the Hedgehog and Wnt signaling pathways (*Logan and Nusse, 2004*; *Arce et al., 2006*; *Varjosalo and Taipale, 2008*; *Whitington et al., 2011*).

## Experimentally testable hypotheses

Our models predict which input TFs are relevant for a given enhancer, and whether they act as activators or repressors. In the case of *eve 2* and *eve 3+7*, we showed that many of these predictions are confirmed by independent experiments already in the literature. These studies involve either perturbing a candidate regulator by mutation, over-expression or misexpression, or mutagenizing binding sites for a candidate regulator in an enhancer sequence, and then measuring the expression of *eve*. To confirm our predictions, we made qualitative comparisons between published data (in the form of a single representative image) and our model predictions. Having validated our modeling framework on these well-characterized enhancers, we can now broadly apply this framework to discover regulators for less well-characterized enhancers in this system. While many enhancers in this network have been mapped by computational studies and functional genomics (*Berman et al., 2002*; *Schroeder et al., 2004*; *Kazemian et al., 2010*; *Négre et al., 2011*; *Schroeder et al., 2011*), our knowledge of most of their regulatory input functions remains incomplete. Our modeling framework complements existing functional genomic and bioinformatics approaches: combined

they will allow a comprehensive description of the relevant inputs of each of these enhancers, and how those inputs work together to produce an output expression pattern.

Our models also point to a role for concentration-dependent effects of Hb and Bcd on their targets. We hypothesize that this is due to concentration-dependent differences in protein-protein interactions, perhaps mediated by the arrangement of TF binding sites in an enhancer, as has been proposed for Hb (*Papatsenko and Levine, 2008*). To test whether binding site arrangements are important, the binding sites for Bcd and Hb can be rearranged within the *eve 2* and *eve 3+7* enhancers, and the output of these mutated enhancers measured. To test which parts of the TFs are involved in mediating protein-protein interactions, the TFs themselves can be mutated, and protein–protein interactions can be assayed by in vitro binding studies. Finally, to test the concentration-dependent effects directly, the concentration of Hb and Bcd can be manipulated in vivo by over-expression, knock-down and misexpression. Our modeling framework is especially useful in this last case, as predictions with and without concentration-dependent effects can be compared. We propose that misexpression studies are likely to be particularly informative, based on the fine-scale differences such as stripe bending and bulging that we were able to predict.

Instead of making qualitative comparisons to experimental data, it would be ideal to test our models quantitatively at cellular resolution. This is possible if we create additional Virtual Embryo data where perturbations, both to input TFs and enhancer sequences, are measured. For knock-down, over-expression or misexpression of TFs, we will need to create a new Virtual Embryo for each perturbation. This will capture all of the direct and indirect consequences of perturbing the TF. We can assess the consequences of mutating enhancer sequences by integrating transgenic reporters into any given Virtual Embryo dataset, as in *Wunderlich et al., 2012*. Creating these new datasets is not a trivial undertaking technically but it would provide the framework for us to directly compare the output of our model predictions to experimental data at cellular resolution to detect fine-scale differences, and without making assumptions about indirect effects. For example, this would allow us to test our proposed role for *tll* in repressing the posterior border of *eve* stripe 7, where classic experiments have been inconclusive and to validate future predictions for other enhancers in the segmentation network. We fully anticipate that analyzing this type of data will lead to further refinements of our models.

## General applicability of our modeling approach

Clearly, our model depends on the quality of the data in the Virtual Embryo, which was derived from many in situ hybridization images of the *Drosophila* blastoderm (*Keränen et al., 2006*; *Luengo Hendriks et al., 2006*; *Fowlkes et al., 2008*). To predict spatiotemporal expression patterns, it's important that the measurements are quantitative and at the resolution of individual cells. One advantage of the blastoderm is that the relevant nuclei are near the surface of the embryo, making it easier to segment the overall fluorescence signal and assign it to individual nuclei. However, microscopy and other techniques such as single-cell transcriptomics are continually improving (*Kalisky et al., 2011*); we anticipate that many comparable datasets will become available over time, both for other developmental time points in *Drosophila*, and in other model systems. Our study demonstrates how theoretical models can be applied to such data in order to make new biological discoveries.

## Methods

### Virtual Embryo dataset

Release 2.0 of the Virtual Embryo dataset was downloaded from the Berkeley *Drosophila* Transcription Network Project website (http://bdtnp.lbl.gov/Fly-Net/) (*Fowlkes et al., 2008*). The release contains composited mRNA expression measurements for 95 genes in 6078 nuclei at six time points (or 'cohorts'). Also provided are protein expression data for four gene products (Bcd, Hb, Kr, and Gt) for some of the time points. Data for the current study were extracted from a 'comma-separated values' (CSV) format Virtual Embryo file (D_mel_wt__atlas_r2.vpc): each row corresponds to a nucleus in the embryo, with columns containing measurements including three-dimensional coordinates, average expression level for a given gene, time point for measurement etc. Expression measurements are provided as relative values for each nucleus, ranging from '0' for minimum expression across all six time points to a little over '1' for maximum expression (e.g.,, the maximum for *eve* is 1.11 and for Hb

it is 1.05). The variability in the maximum is a result of the method used to determine the relative variation between nuclei across different time points in the Virtual Embryo (*Fowlkes et al., 2008*).

The coordinates of the Virtual Embryo are along the anteroposterior (x), left-right (y) and dorso-ventral (z) axes. The difference between the minimum and maximum is 404 μm for the x-coordinate, 154 μm for the y-coordinate and 155 μm for the z-coordinate.

## Training data preparation

Training was performed using expression measurements at the third time point (Cohort 3). 6,078 nuclei were classed as ON (2444) or OFF (3634) depending on whether *eve*'s expression is above or below the threshold of 0.2 (approximately 20% of maximum). Nuclei were grouped into the seven *eve* stripes making use of the neighboring nuclei information provided in the *Virtual Embryo* (stripe 2 = 348 nuclei, stripe 3 = 342 nuclei, stripe 7 = 383 nuclei). mRNA expression measurements for 34 genes were included in the training data (*brk, bun, cad, CG10924, CG17786, CG4702, cnc, croc, Cyp310a1, D, Dfd, Doc2, emc, fj, fkh, hkb, kni, knrl, oc, path, rho, sala, slp1, slp2, sna, sob, srp, term, tll, Traf1, trn, tsh, twi, zen*). For four TFs (Bcd, Hb, Kr, Gt), we used the protein expression measurements instead.

## Training logistic regression models

Logistic regression was used to model *eve* expression by linking the regulator concentrations as continuous input variables, to *eve*'s expression state as the binary output. For a nucleus *i*, the predictor, $\eta_i$, is calculated as a linear combination of concentrations:

$$\eta_i = \beta_0 + \beta_1 x_{1i} + \ldots + \beta_k x_{ki}$$

where $x_{ki}$ is the expression measurement of the *k*th gene for the *i*th nucleus with the $\beta$ to be estimated. For the quadratic models, a single quadratic term was added for the regulator, q, in question:

$$\eta_i = \beta_0 + \beta_1 x_{1i} + \ldots + \beta_k x_{ki} + \beta_{k+1} x_{qi}^2$$

The predictor is linked to the estimated probability $p_i$ of *eve* being ON in the *i*th nucleus:

$$p_i = \frac{1}{1 + e^{-\eta_i}}$$

Models for *eve 2* were trained using the 348 nuclei defined as ON in stripe 2, as well as the nuclei defined as OFF excluding the nuclei of other stripes and their immediate neighbors (2588 nuclei). Similarly, models for *eve 3+7* were trained using 725 ON nuclei in stripes 3 and 7, and 2756 OFF nuclei.

The models were fitted using the R function glm from the stats package, which uses Iteratively Re-weighted Least Squares. For our best fitting models, glm issued a fitting and evaluation warning message. This was because most of the logistic models that classify the *eve* stripes successfully have some fitted probabilities very near 0 or 1. (The nuclei on the borders of the stripes have intermediate values). Although this can suggest problems in certain situations, here, in agreement with Ripley (*Ripley, 2008*) it is viewed as a desirable outcome of classification. The trained model was then used to predict *eve* expression in all 6078 nuclei across the entire embryo, using the concentrations of the relevant regulators.

## Tests of model consistency across different training subsets

The consistency of the model across different training subsets was tested in several ways. (i) Each model was trained on a subset of the training dataset and then used to predict *eve* expression for the whole embryo. Subsets used included: nuclei within 20 μm either side of the lateral midline; nuclei within the relevant stripe(s) and only their immediate neighbors; and a cross-validation test, which was the average of 100 predictions each trained on a random subset of 50 nuclei. For *eve 3+7*, two extra subsets excluded the ON nuclei from either stripe 3 or 7. Less consistent models produce poor predictions after training on some subsets. (ii) Each model was trained using all 38 regulators and then used to predict *eve* expression for the whole embryo. Models suffering from localized over-fitting show fragmented *eve* expression. (iii) Models were trained for each of the stripes in turn, using the regulators of the best-fitting models (such as Bcd, Hb, Gt, and Kr for *eve 2*). This showed that the given regulators are not able to fit any arbitrary stripe well.

## Predictions of regulatory perturbations

The effects of regulatory perturbations were simulated by adjusting the concentrations of the relevant regulator without changing any model parameters (i.e., without retraining), and then predicting *eve* expression across the whole embryo. Binding site mutations and null mutants were simulated by setting the regulator concentration to '0' in all nuclei. Where indicated, indirect effects were simulated by adjusting the expression level of downstream regulators and again, predicting *eve* expression without any model adjustments. Other types of regulatory perturbations, such as the misexpression studies, were performed similarly by adjusting regulator concentrations as described in the main text.

## Visual display of model outputs

Model predictions of wild-type and mutant *eve* expression are displayed graphically for each nucleus in the embryo. The Virtual Embryo contains three-dimensional coordinates for each nucleus, making it possible to show the predictions in their spatial context. In most figures, embryos are shown from two perspectives: lateral and three-dimensional. In the lateral perspective, each nucleus is plotted using the $(x,z)$ coordinate, ignoring the $y$ coordinate. The $x$- and $z$-axes are aligned to the anteroposterior (left to right) and dorsoventral (top to bottom) axes respectively, so showing a view from the left side of the embryo. Since predictions for the left and right sides are similar, all nuclei (i.e., both left and right) are plotted in one composite view from the left side of the embryo. The three-dimensional perspective is plotted using the cloud function from the lattice package in R, similarly from an anterior perspective. Nuclei are colored according to the model's prediction, from $p=0$ (light) to $p=1$ (dark). The color scale for predictions within stripes is grey-scale and predictions outside of stripes are shown on a red scale, with peach for values below 0.15.

## Calculating the accuracy of model outputs

To accompany the visual display of wild-type predictions, we also calculated percentage accuracies to aid comparison between alternative models. These values provide good indications of model performance in predicting the stripe boundaries. For each model, we calculated the proportion of nuclei predicted as being ON ($p>0.5$) within the stripe(s) under consideration (i.e., true positives), in nuclei immediately adjacent to the stripe nuclei, and two nuclei away (i.e., false positives). The identities of neighboring nuclei are provided by the Virtual Embryo dataset.

## Regulator discovery

For *eve 2*, we trained all possible linear models using four out of 38 regulators in the dataset (total 73,815 models), using the log likelihood of each fitted model as its score. A similar approach was used for exploring quadratic models, except that any model containing Bcd and/or Hb also included the corresponding quadratic term(s). The results are summarized as heat maps as shown in *Figure 3*.

## Software

Analysis was performed with R version 2.15.1 (*R Core Team, 2012*), using colors from the ColorBrewer palettes in the RColorBrewer package. Plots made use of the lattice, ggplot2 and RBGL packages. The graph package was used to select neighboring nuclei.

# Acknowledgements

We would like to thank the members of the DePace laboratory, in particular Tara Lydiard-Martin, Max V Staller, Zeba Wunderlich, and Ben Vincent, for insightful discussions throughout the project.

# Additional information

### Funding

| Funder | Author |
| --- | --- |
| EMBL | Garth R Ilsley, Rolf Apweiler, Nicholas M Luscombe |
| Cancer Research UK | Nicholas M Luscombe |
| Okinawa Institute of Science and Technology | Garth R Ilsley, Nicholas M Luscombe |

| Funder | Author |
|---|---|
| National Institutes of Health | Rolf Apweiler, Angela H DePace |
| Microsoft Research | Jasmin Fisher |
| Peterhouse, Cambridge | Garth R Ilsley |
| University College London | Nicholas M Luscombe |

The funders had no role in study design, data collection and interpretation, or the decision to submit the work for publication.

## Author contributions

GRI, Selection and preparation of data, Conception and design, Analysis and interpretation of data, Drafting or revising the article; JF, RA, Supervisory role, Discussions, Drafting or revising the article; AHD, NML, Supervisory role, Discussions, Interpretation of data, Drafting or revising the article

## Additonal files

### Major dataset

The following previously published dataset was used:

| Author(s) | Year | Dataset title | Dataset ID and/or URL | Database, license, and accessibility information |
|---|---|---|---|---|
| Fowlkes CC, Hendriks CL, Keränen SV, Weber GH, Rübel O, Huang MY, et al. | 2008 | A quantitative spatiotemporal atlas of gene expression in the *Drosophila* blastoderm | http://bdtnp.lbl.gov/Fly-Net/bidatlas.jsp | Publicly available at http://bdtnp.lbl.gov/. |

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
