## [Decision Letter]

Thank you for sending your work entitled “Cellular resolution models for *even skipped* regulation in the entire *Drosophila* embryo” for consideration at *eLife*. Your article has been favorably evaluated by a Senior editor and 3 reviewers, one of whom is a member of our Board of Reviewing Editors.

The following individuals responsible for the peer review of your submission each wish to reveal their identity: Roderic Guigo (Reviewing editor); Mike Levine (peer reviewer).

The Reviewing editor and the other reviewers discussed their comments before we reached this decision, and the Reviewing editor has assembled the following comments to help you prepare a revised submission.

1) Two of the reviewers raised concerns about the statistical methods employed to validate the model. Specifically, one of the reviewers stated that the conclusions should be drawn from a blind test or N-fold cross validation; however, the authors used this to test the model over-fitting as a separate section. Nevertheless, if we understood them correctly, the main conclusions presented are drawn from using the entire set of data points, including those used for training. This is circular, and the authors should clarify this point in the main text. The over-fitting assessment is not generally necessary here, as the number of parameters is considerably below the number of independent data points.

2) Two of the reviewers also suggested that the authors should further investigate the correlation between the input variables and test models including fewer variables. One of the reviewers suggested that you might use stepwise logistic regression as a way to select those variables that are truly informative. Using all combinations of 4 variables (in the case of stripe 2) did not seem the optimal way to infer the minimum combination of maximally informative variables.

3) Two of the reviewers also raised concerns about the way you attempted to simulate the effects of perturbation by manipulating the input signal of the regression model without changing its learned coefficients. One of the referees specifically believes that this practice is incorrect, simply because the model was learned and optimized based on 3 input signal, and should be re-optimized if you decided to remove one input. Again the model comparison should be conducted by a blind test, not all entire dataset used for training.

4) Two of the reviewers asked themselves whether it would not be of interest to test the stability of the models during development. One of the referees specifically encouraged the authors to extend their models to earlier stages of development, namely the first 20 min of nuclear cleavage cycle 14.

5) The utility of the models rely on their capacity to generate testable hypotheses. In this regard, one of the referees asked whether any novel prediction that could be experimentally tested has been derived from the model. This is an issue that should be made more explicit by the authors.

---

## [Author Response]

*1) Two of the reviewers raised concerns about the statistical methods employed to validate the model. Specifically, one of the reviewers stated that the conclusions should be drawn from a blind test or N-fold cross validation; however, the authors used this to test the model over-fitting as a separate section. Nevertheless, if we understood them correctly, the main conclusions presented are drawn from using the entire set of data points, including those used for training. This is circular, and the authors should clarify this point in the main text. The over-fitting assessment is not generally necessary here, as the number of parameters is considerably below the number of independent data points*.

We thank the reviewers for their comments. Briefly, two reviewers suggest that the conclusions of the study should not be based on how well our model fits the training data (and hence recommend cross-validation or a blind test). Their concern is that this is circular, but whether it is or not depends on the conclusions being drawn. In this matter, it is perhaps best to understand our work as a regression model that describes the relationship observed between variables in a dataset—the Virtual Embryo—rather than as a machine-learning classifier whose expected performance on other data needs to be assessed in an unbiased way. For this reason, the first part of our work draws conclusions based on the fit to the data.

These data include the measurement of expression in the early embryo in the *eve* stripe of interest as well as in the nuclei that are outside any stripe. Our most important conclusion is that *eve* expression in these nuclei can be correctly separated into two classes using the measured concentrations of various transcription factors alone. This is biologically interesting and relevant—for instance, in its implications for positional information in the early embryo. Perhaps what is confusing is that in justifying these conclusions we make use of predictions across the whole embryo, rather than just on the training data. However, the intention here is not to test the generality of the model or its validity as a classifier of independent data, but rather to assess the applicability of the model across the whole embryo; that is, whether the information content in the other stripes is consistent with the current model (or, biologically-speaking, whether a different source of information would be necessary to control activation of the enhancer in these regions e.g. additional signaling, epigenetic state, etc). We have now clarified this in the main text.

We agree that an over-fitting assessment is not generally necessary, but nevertheless, it is useful to ask whether the statistical relationship expressed in the model is consistent across different subsets of the data. This was the primary point of the separate section where we presented the results of the random subsampling cross-validation test, as well as the tests where we considered other subsets, e.g., a narrow strip along the dorsoventral axis and the stripe and its neighbouring nuclei. Further, although our model up to this point in the manuscript demonstrates that transcription factor concentrations alone provide sufficient positional information, we were interested in whether the ability of our model to fit the data might be constrained in biologically relevant ways. It was for this reason that we also tested whether our model, with the known regulators, could fit any stripe. We presented these findings under the heading of “over-fitting”, but in light of the reviewers’ comments we clarified this in the section renamed “The model performs consistently across different subsets of the data”.

Going beyond these conclusions, we then make use of independent verifications, such as a comparison of our models’ predictions with mutant and misexpression data. In addition, in response to a reviewer’s request, we have added the predictions of our models at earlier time points. Although these data are from the BDTNP, they were not used during our models’ development, and hence this can be considered an independent test.

*2) Two of the reviewers also suggested that the authors should further investigate the correlation between the input variables and test models including fewer variables. One of the reviewers suggested that you might use stepwise logistic regression as a way to select those variables that are truly informative. Using all combinations of 4 variables (in the case of stripe 2) did not seem the optimal way to infer the minimum combination of maximally informative variables*.

We did, in fact, explore stepwise logistic regression during model development, although these results were not reported in the original paper. We found that stepwise selection was a successful procedure for finding putative regulators, but that it generally includes more regulators than necessary for a good visual fit (for example, a stepwise selection procedure for a linear model of *eve 3+7* with the Bayesian Information Criterion finds 15 regulators). The stopping point (i.e., the penalty for adding an extra parameter) is effectively arbitrary in this case, or at least difficult to determine *a priori* in a justifiable manner. Since we are interested in a minimal model that can explain the observed expression and found that four regulators is sufficient, we could put an upper bound on the number of regulators to consider in our models, at least as a starting point.

In practice, though, we are also interested in models that contain the key regulators, and in the case of *eve 2*, there are four. So, for example, the best scoring model for *eve 2* with three regulators (Hb, Gt and Bcd) can produce a reasonable although less sharp fit than when Kr is included, but making use of this model would restrict our ability to consider *Kr* mutant data. For *eve 3+7*, the best scoring linear model with three regulators (Gt, kni and tll) does not include Hb, an important regulator. It is also worth noting that these three regulators are the same as those picked up in our regulatory discovery method (Figure 4—figure supplement 6).

Further, the goal of our discovery method was not simply to pick the best four (or three) regulator models (although these are reasonable). Instead, we are interested to see which regulators work together to control the spatial pattern observed, such as when two repressors define the borders of a stripe. In this case, it is valuable to consider how a transcription factor performs in the context of other transcription factors. This is not taken into account in stepwise selection. Instead, our approach rigorously compares the contribution of each pair of transcription factors in the context of the most informative two-regulator model from the remaining regulators. This, as we show, clearly highlights informative regulators, which are indeed able to predict expression in the early embryo. Nevertheless, we do not claim to have found all informative regulators, but rather propose a minimal model that works under the conditions analysed.

*3) Two of the reviewers also raised concerns about the way you attempted to simulate the effects of perturbation by manipulating the input signal of the regression model without changing its learned coefficients. One of the referees specifically believes that this practice is incorrect, simply because the model was learned and optimized based on 3 input signal, and should be re-optimized if you decided to remove one input. Again the model comparison should be conducted by a blind test, not all entire dataset used for training*.

Briefly, two reviewers suggest that if we wish to manipulate the input signals (i.e., in the perturbation tests), we should re-optimise the parameters under these new conditions. We agree this would give the models a fairer chance of classifying the data points correctly; this would be particularly important if our goal was to consider how a model with fewer inputs might perform, or how the classification approach might work with different data.

However, at this point in our analysis we have already accepted the models as plausible, and we are rather considering the hypothesis that the selected models represent the underlying biological processes. The purpose in modifying the inputs to these models is to compare their predictions with the results of experimental perturbations. In these experiments the intention is to modify one of the molecular inputs to the enhancer while leaving the others unchanged. For this reason, to compare like with like, it is important that we do not change the other parameters of the models, and hence we do not to re-optimise them.

*4) Two of the reviewers asked themselves whether it would not be of interest to test the stability of the models during development. One of the referees specifically encouraged the authors to extend their models to earlier stages of development, namely the first 20 min of nuclear cleavage cycle 14*.

We thank the reviewers for this suggestion. The Virtual Embryo dataset comprises 6 time points during nuclear cleavage cycle 14, each about 10 minutes apart. Most of the presented work was performed using data from time point 3, when *eve* expression becomes sharply defined. To address the reviewers’ suggestion, we selected the best performing models for *eve 2* and eve *3+7* trained on time point 3, and applied them to data from time points 1 and 2. As shown in the new Figure 6 and described in the section “Models predict *eve 2* and *3+7* expression in early time points”, the models successfully reproduce the stripe formation apparent in the Virtual Embryo; the outputs are also consistent with previously published *in situ* hybridisations.

*5) The utility of the models rely on their capacity to generate testable hypotheses. In this regard, one of the referees asked whether any novel prediction that could be experimentally tested has been derived from the model. This is an issue that should be made more explicit by the authors*.

We have now included a modified section (“Experimentally testable hypotheses”) describing: (i) specific experiments to clarify which regulatory mechanisms apply for the *eve 2* and *eve 3+7* enhancers (e.g., linear vs quadratic models); and (ii) further general experimental approaches that would advance our understanding of gene regulation in early fly embryos.